# Internal Validation of the ASFV MONODOSE dtec-qPCR Kit for African Swine Fever Virus Detection under the UNE-EN ISO/IEC 17025:2005 Criteria

**DOI:** 10.3390/vetsci10090564

**Published:** 2023-09-08

**Authors:** Gema Bru, Marta Martínez-Candela, Paloma Romero, Aaron Navarro, Antonio Martínez-Murcia

**Affiliations:** 1Genetic PCR Solutions™, 03300 Orihuela, Spainresearch@geneticpcr.com (A.N.); 2Department of Microbiology, University Miguel Hernández, 03312 Orihuela, Spain

**Keywords:** ASFV, diagnosis, ISO/IEC 17025:2005, qPCR, validation

## Abstract

**Simple Summary:**

African swine fever virus is a pathogen capable of spreading among swine populations, which is usually fatal since no vaccines are currently available on the market. Outbreaks caused by this virus are the reason for massive economic losses on pig farms and are a matter of worry for the swine sector worldwide. Effective and reliable detection of the virus is relevant to prevent uncontrolled contagion, especially when no effective preventive measures can be applied. In order to confront the spread of African swine fever, several rapid real-time polymerase chain reaction detection assays were developed by different research groups. In the present study, we focused on the validation of the ASFV MONODOSE dtec-qPCR kit using reference genetic material provided by national and international reference laboratories, and results were compared with those obtained from using internationally accepted reference methods. The data obtained indicated that the kit can be used in veterinary samples. Technical innovations enable a user-friendly and rapid management of samples, reducing the probability of human errors happening and transport costs.

**Abstract:**

African swine fever virus is considered an emerging virus that causes African swine fever, a disease characterised by high mortality and elevated transmission rates and that, as it is for most other viral diseases, cannot be treated with specific drugs. Effective and reliable detection of the virus is relevant to prevent uncontrolled contagion among boar populations and to reduce economic losses. Moreover, animal health laboratories are demanding standardisation, optimisation and quality assurance of the available diagnostic assays. In the present study, the ASFV MONODOSE dtec-qPCR kit was validated following the UNE-EN ISO/IEC 17025:2005 guidelines. Analytical validation terms include in silico and in vitro specificity, sensitivity, efficiency and reliability (repeatability/reproducibility). Diagnostic validation of the method was assessed through the analysis of a total of 181 porcine samples originating from six different matrix types doped with African swine fever virus DNA received from the European reference laboratory for African Swine Fever (INIA-CISA, Madrid, Spain): whole blood, blood serum, kidney, heart, liver and tonsil. Results agreed with those obtained from a reference detection method also based on real-time PCR, endorsed by WOAH, but the ASFV MONODOSE dtec-qPCR kit incorporates some technical innovations and improvements which may benefit end-users. This kit, available worldwide with full analytical and diagnostic validation, can recognise all known ASFV genotypes and brings additional benefits to the current qPCR technology.

## 1. Introduction

African swine fever virus (ASFV) is a large, enveloped, double-stranded DNA virus, and it is the sole member of the genus *Asfarvirus*, the only member within the family Asfarviridae [1]. ASFV is the causative agent of African swine fever (ASF), a transboundary swine viral disease which affects the Suidae family, and it is responsible for increased mortality rates in pig farms worldwide, generating one of the main crises in the swine industry [2,3]. According to genomes sequenced and deposited in public databases, ASFV revealed a great diversity and, so far, 24 genotypes have been described on the basis of partial sequencing of the p72 protein-coding region included in the B646L gene [4,5,6]. Genotyping of viral isolates may help identify the origin of an outbreak, providing an insight into the viral distribution and route of transmission [4,7].

Regarding ASFV transmission, eight species of the *Ornithodoros* genus of soft ticks have been identified as viral vectors and reservoir hosts. In Africa, the virus is maintained though a sylvatic cycle between soft ticks and warthogs; nevertheless, some tick species present in Africa and Europe are capable of infecting domestic swine [8]. Wild boars are, at least, as susceptible to this virus as domestic pigs, but the disease has established self-sustaining cycles in wild swine, helping ASF spread and maintain its prevalence [9]. In Europe and Asia, transmission between pigs and boars can happen when livestock–wildlife interaction occurs on farms with deficient biosecurity [8]. Reports also indicate a large number of cases in the wild boar population, which is consistently increasing and spreading across large geographic areas, therefore representing a virus reservoir dangerous for the pig industry [8,9]. ASFV is highly resistant to extreme conditions, as it can survive under diverse environmental circumstances and may be transmitted through contaminated objects [10]. This malignancy does not only harm the health and welfare of animals, but also has a negative impact on biodiversity and causes important financial losses for producers [3]. The considerable mortality rate, rapid spread and negative socioeconomic impact are some of the reasons why the notification of ASF to the World Organization for Animal Health (WOAH, formerly OIE) is mandatory [11,12]. It seems necessary to continue improving current methodologies to identify ASF-affected animals in order to enhance the prevention of viral escalation and to reduce economic losses caused by the expansion of the pandemic [11,13]. Though the identification of infected individuals, areas where the virus has been eradicated will continue to be free of ASFV by preventing the introduction of the disease [13].

Laboratory molecular diagnostic techniques for infectious agents need standardisation, optimisation and quality assurance, as demanded by national and international authorities, which require objective evidence that the diagnostic assays are as reliable as possible [14,15]. ASFV detection methods recommended by the WOAH are virus isolation, antigen measurement using fluorescent antibody tests (FAT) and viral genome detection via polymerase chain reaction (PCR). Virus isolation is considered the gold standard diagnostic technique for ASFV; however, it is time-consuming and labour-intensive [16,17]. FAT are a methodology that have been used to detect ASFV antigens in pig meat, and, even though it is a quite sensitive method when analysing peracute and acute ASF cases, the sensitivity decreases up to 40% in subacute and chronic disease due to the formation of antigen–antibody complexes in the tissues of infected swine [18]. Currently, the use of PCR-based methods for the detection of ASFV is widespread, since they have been proved to be sensitive, rapid and specific alternatives to virus isolation and antigen detection. PCR assays enable virus detection even in inactivated samples [19].

Even though effective PCR [20,21] and real-time PCR (qPCR) [22,23,24,25,26] assays to detect ASFV have been previously developed, there are still opportunities for further enhancement in order to fulfil some additional requests from diagnostic laboratories. The validation of the ASFV MONODOSE dtec-qPCR kit (GPS™, Orihuela, Spain) could offer the possibility of a fast and reliable method to detect all known genotypes of ASFV. This kit can be easily distributed worldwide without substantial costs, since it can be transported at room temperature in the absence of dried-ice. The newly developed MONODOSE format, in which all the reagents are included in individual ready-to-use tubes, enables technicians to rapidly prepare the mix for testing. The fast thermic cycling allows for the qPCR time to take about half an hour. In the present report, the results from the analytical and diagnostic validation of the qPCR-kit for ASFV detection following the UNE-EN ISO/IEC 17025:2005 criteria [27] are presented and compared with two reference qPCR methods [22,26] suggested by the WOAH.

## 2. Materials and Methods

### 2.1. DNA Extraction/Purification

The genetic material of all samples was extracted and purified with a silica column-based kit GPSpin Viral DNA/RNA (GPS™, Orihuela, Spain). The extraction and purification of total DNA was performed using 200 µL of the material studied, following the instructions provided by the manufacturer. With the objective of evaluating the extraction yield of the method, DNase/RNase-free water (molecular grade) was doped with 5 µL of ASFV genetic material. Additionally, to test the robustness of the method for the extraction and purification of ASFV from porcine samples, six different matrices (whole blood, blood serum, kidney, heart, liver and tonsil) where doped with 5 µL of reference total DNA. In the case of the viscera (kidney, heart, liver and tonsil), a previous sample-homogenisation step was carried out with a 0.9% NaCl solution. The yield and robustness of extraction from the doped samples were estimated by comparing the resulting copy number to that initially obtained for each reference genome.

### 2.2. qPCR Protocol

The term MONODOSE refers to a commercial kit-format which only requires the sample to be added to individual PCR tubes already containing all the required reagents for the reaction to occur; the protocol described by the manufacturer was followed [28]. During the development of the present study, 5 µL of each sample analysed was added to the supplied ready-to-use tubes. The qPCR reactions were carried out in a QuantStudio3 (ABI) device, with a thermal protocol composed of an activation step at 95 °C for 2 min and 40 amplification cycles organised as follows: denaturation at 95 °C for 5 s, annealing/extension at 60 °C for 20 s and, in the end, data collection. Positive control using the Standard Template, negative control (nuclease-free water) and internal control (IC) were included in all tests, with all these materials supplied in the qPCR kit. Data were collected using the HEX channel to read the IC and using the FAM channel for the target. The IC was optimised to detect the presence of possible qPCR inhibitors in the case of negative samples without affecting the main target sensitivity. It was designed against an external target not related to ASFV or to the host. Calibration of the qPCR was performed with a standard curve prepared from ten-fold dilution series of a Standard Template provided in the kit, with a range between 10^6^ and 10 copies.

For the qPCR method developed by Fernández-Pinero et al. [26] for the molecular diagnosis of ASFV by using Universal Probe Library, oligonucleotides were purchased from Integrated DNA Technologies (IDT, Coralville, IA, USA). The thermal protocol included an activation step of 5 min at 95 °C and 45 amplification cycles, which encompassed the denaturation step at 95 °C for 10 s and the annealing/extension step at 60 °C for 30 s. Fluorescence acquisition was in the FAM channel. The PCR assay developed by King et al. [22] was adapted to be used as a qPCR test as described by the WOAH. The thermal protocol started with an activation step of 5 min at 95 °C and was followed by 45 amplification cycles consisting of denaturation at 95 °C for 10 s and annealing/extension at 58 °C for 30 s.

### 2.3. Analytical Specificity

Analytical specificity, as the ability of the method to recognise only the target sequence but no other homologous sequences belonging to different related species, was assessed in silico and in vitro. During the designing phase, in silico analytical specificity was examined using the BLAST + 2.14.0: 25 April 2023 software, publicly available at the online site of the National Center for Biotechnology Information (NCBI) (Bethesda, MD, USA) [29]. Alignments were performed against all strain sequences of the target species to ensure complete inclusivity; these sequence alignments were carried out using the Mega 5.2.2 software [30]. ASFV primers and probe sequences showed a notably high degree of identity to the target present in all ASFV entries available on the NCBI public database (up to May 2023, more than 1700 target sequences belonging to the 24 known genotypes) [29]. Exclusivity in silico was analysed considering the sequences from all taxa described and deposited in public databases. A deeper evaluation of viral species which share a common host with ASFV was performed for Classical swine fever virus (CSFV), Porcine respiratory and reproductive syndrome virus (PRRSV), Porcine circovirus types I and II (PCV-I and PCV-II), Suid herpesvirus 1 (SuHV-1), Foot-and-mouth disease virus (FMDV), Swine vesicular disease virus (SVDV) and Vesicular stomatitis Indiana virus (VSIV).

The qPCR was tested in vitro using the genetic material of 21 ASFV reference DNAs from the European reference laboratory for African Swine Fever: Centro de Investigación en Sanidad Animal (INIA-CISA) (Madrid, Spain). These 21 reference total DNAs were representative of 7 of the 24 virus genotypes. This study was completed by testing 8 reference ASFV samples at the Veterinary Hygiene Department from Warsaw, Poland (PIWet).

### 2.4. Analytical Sensitivity

The assay to detect ASFV though qPCR was subjected to careful validation according to the guidelines of the international norm UNE-EN ISO/IEC 17025:2005. The following parameters were estimated: standard curve calibration and linearity analysis (against ten-fold serial dilutions of 10^6^ to 10 copies of Standard Template provided in the kit), reliability (repeatability and reproducibility), limit of detection (LOD) and limit of quantification (LOQ). With the objective of obtaining statistically relevant results, all the variables listed above were evaluated with a minimum of 10 replicates as previously described [28,31].

### 2.5. Diagnostic Performance

The diagnostic specificity and sensitivity were assessed by testing 181 samples of porcine origin from six different matrix types (whole blood, blood serum, kidney, heart, liver and tonsil). A total of 105 samples were prepared (doped) with 5 µL of each ASFV reference genome from INIA-CISA and 76 remained unmodified (not doped). Two reference detection methods based on qPCR [22,26] were used to test samples and to assess the diagnostic sensitivity and specificity of the kit under evaluation.

## 3. Results

When the kit was subjected to the in silico inclusivity analysis through alignments between the primers/probe sequences and all ASFV sequences available in public databases (1701 target sequences up to May 2023), the results showed that the qPCR design was inclusive for all of them. The results obtained from the assessment of the exclusivity in silico showed that alignments performed against sequences available on the database did not match any sequence (not shown). This indicated that no other organism should be detected when performing a qPCR using the reagents of this previously developed kit. Inclusivity in vitro was evaluated with the genetic material of 29 ASFV total DNA from reference laboratories, obtaining positive results for all analysed sequences.

Analytical sensitivity was evaluated through several parameters, and each assay was repeated at least 10 times (15 in the case of LOD and LOQ). The results were evaluated with respect to established criteria for acceptance. The qPCR standard curve used for assay calibration was performed using a 10-fold dilution series containing the 10^6^ to 10 copies of Standard Template provided in the kit and following the manual of the manufacturer (Figure 1).

Calibration of the qPCR test showed Ct values within the interval of 17 and 34. To assess the lineal regression, the slope (a) and coefficient (r2) were calculated; the obtained results were −3.369 and 1.000, respectively. These values were consistent with the criteria for acceptance. A Fisher test with a 95% confidence interval was carried out in order to validate the linear model. The value of the Fassay obtained was lower than the value of Ffisher (3.261 and 5.318, respectively); therefore, the linear model was validated. Efficiency was 98.1% and, thus, accepted since it was found within the established ranges. In conclusion, all the above-mentioned parameters met the criteria of acceptance. For the LOD and LOQ results to be acceptable, coefficient values (CV) should be smaller than 10% (<10%): CV ranged from 0.62 to 1.94% for repeatability, and from 0.86 to 1.65% for reproducibility. The LOD for 10 genomic copies was 100% reproducible and the accuracy of the LOQ for 10 genomic copies was accepted, as the t value (1.161) was lower than the theoretical value from a Student table (tstudent = 2.145). All these parameters and the respective values obtained are summarised in Table 1.

With the objective of evaluating analytical specificity, the in vitro inclusivity of the method was assessed using the genetic material received from the INIA-CISA. The 21 ASFV reference DNAs, representative of seven different genotypes, and eight additional reference samples from Warsaw yielded positive results when the ASFV dtec-qPCR kit was employed for detection (Table 2).

In order to calculate the yield of the extraction method used, a DNase/RNase-free water (molecular grade) extraction was performed before extracting the DNA from doped matrices. The DNA extraction yield (% ± standard deviation, SD) achieved using water samples doped with ASFV genomic DNA was 63.4 ± 6.4%. Moreover, the 21 ASFV reference genomic DNAs were employed to dope the different porcine tissue matrices and to estimate the robustness. The minimum yield was obtained in blood serum samples, with 45.2 ± 12.1%, followed by 69.0 ± 25.7% in kidney samples and 73.9 ± 25.4%, 77.1 ± 21.9% and 79.5 ± 29.6% for liver, heart and tonsil samples, respectively (Figure 2).

Diagnostic validation was carried out using the ASFV MONODOSE dtec-qPCR kit and results were compared with those obtained from two reference qPCR-based methods [22,26]. Of 181 total samples (Appendix A), 105 were doped with viral DNA of ASFV and 76 were left unmodified (not doped). The outcomes of both the GPS™ kit and the reference qPCR method described in Fernández-Pinero et al. [26] were in full agreement, showing a total of 105 positive and 76 negative specimens. However, results obtained with the PCR developed by King et al. [22] yielded 98 positives and 83 negatives, indicating seven discrepancies with respect to the other tests. The matrices in which the obtained results did not coincide with the data of the other tested methods were blood serum, heart, liver and tonsil (Appendix A).

Consequently, when the results from Fernández-Pinero et al. [26] were considered as the reference, the qPCR kit under validation in the present study showed 100% in both parameters, diagnostic specificity and sensitivity, for all the samples tested during this validation. In both cases, the resulting values passed the acceptance criteria of 90%. Taking into account the results obtained for the diagnostic specificity and sensitivity, the diagnostic efficiency was 100%.

## 4. Discussion

ASFV is the pathogen responsible for ASF, one of the most threatening diseases affecting swine, among which are included wild boars and domestic pigs, while being asymptomatic in natural reservoir hosts [32]. Since ASFV can endure extreme environmental conditions, it can be easily transmitted and rapidly spread [8]. The relevance of the disease has led to the redaction of the Council Directive 2002/60/EC (EC, 2002) (European Commission, 2002), which establishes the measures to be applied for ASFV control in the European Union [33]. There are several research groups developing new and diverse technologies to stop the spread of ASF, but, so far, a completely safe and effective vaccine against ASFV is not available on the market [3]. Due to the lack of vaccination, the control of ASF outbreaks depends only on the application of health measures, such as slaughtering infected or suspected-of-being-infected animals, and the cleaning, disinfection and disinfestation of premises, vehicles and equipment [34]. All these circumstances are reasons to remark on the need to develop, validate and make widely accessible a rapid, simple and accurate method for ASFV detection to prevent virus spread and to reduce economic losses.

GPS™ has developed a new real-time PCR for ASFV detection which improves the manageability of the qPCR technology and minimises the time and cost required to perform the tests. This kit has been subjected to validation following the recommendations indicated by the international norm ISO/IEC 17025:2005 and compared with reference methods. The results obtained from the validation of the kit were optimum according to the ranges established as acceptance criteria derived from the guidelines of the norm (Table 1). Specificity in silico was assessed through primer and probe sequence analysis, while specificity in vitro was studied with reference genomic material. The results seemed to indicate that the newly developed methodology is inclusive for all genotypes of ASFV described so far. In addition, the in silico exclusivity of the design was proven by analysing all sequences available in public databases and, in particular, those viral species that share a natural host with the target and may be present in analysed samples. The evaluation of the quantitative PCR phase inside the range of the standard curve is essential to perform quantification, and, for that purpose, it was examined whether the average efficiency and R2 values obtained were acceptable in the ranges established by the norm. For the validation process, it was considered appropriate to replicate the analysis a minimum of 10 times. The Fisher test was applied to evaluate the linear model, and it was found acceptable since Fassay was significantly below Ffisher. The performance of the method can be evaluated using the efficiency (e), which was found to be 98.1%; although e values above 75% are usually considered acceptable, results above 90% will ensure the yield of the PCR amplification is optimum. The method was found to be repeatable and reproducible because the coefficient of variation (CV) was lower than 10% in all cases, hence being considered reliable. Sensitivity was assessed during the validation of the method since it indicates the ability of the kit to correctly detect the target. The current study provided positive outcomes in 100% of the cases where the detection of 10 copies was tested to determine if the limit of detection (LOD) was reproducible. Finally, the limit of quantification (LOQ) result was accepted, as it was within the stipulated range because the t value obtained was lower than the tstudent value (1.161 < 2.145). The values obtained for the diagnostic validation of the samples analysed showed a 100% diagnostic specificity and 100% diagnostic sensitivity when Fernández-Pinero et al. [26] was taken as the reference method for this validation. In conclusion, the results of the analysed parameters passed the acceptance criteria established for the validation and agreed with the data obtained from the most recently developed qPCR method recommended by the WOAH, verifying the method to be suitable for use (Table 1).

Between different ASFV detection methods recommended by the WOAH, conventional and qPCR assays have been adopted for routine diagnosis in reference laboratories [20,22,26]. Out of the methodologies recommended for ASFV detection in the WOAH terrestrial manual, it was decided to contrast the performance of the ASFV MONODOSE dtec-qPCR detection kit only with the procedures that could be used to perform qPCR assays [16]. The results obtained with the kit under validation in the present study and the method developed by Fernández-Pinero et al. [26] showed the same outcome for both methodologies. However, the results obtained with the qPCR by King et al. [22] presented discrepancies in seven specimens, which in this case resulted negative (Appendix A). The highest number of discordant results was obtained in blood serum, which could be related to the lower extraction yield observed for that matrix during the evaluation of the robustness of the chosen method of extraction. Nevertheless, discrepancies were also obtained in heart, liver and tonsil doped samples, which showed very favourable results regarding extraction robustness. In all, these data suggest that a correlation between the matrices employed and the results obtained cannot be established. Another possibility was that these discrepancies could be due to the sequence specificity of the primers and probe of King’s method [22]. At the time of designing that qPCR design, the number of ASFV sequences available in public repositories was probably low. Currently, there are 249 complete genome sequences in the NCBI database which belong to ASFV; meanwhile, in 2003 there were only 6 complete genome sequences uploaded. This may be considered an indicator of the ongoing growth of information available in public databases. Newer studies can access a higher number of widely accessible sequences than the reference methods established decades prior. This illustrates the relevance of performing periodical updates to the specificity of current qPCR assays. Furthermore, a report published prior to this study [35] showed that the WOAH-validated PCR method developed by King et al. [22] had a low sensitivity, since it was unable to detect the virus in a sample obtained from an asymptomatic pig experimentally exposed to the virus and also failed to detect three samples obtained from hunted wild boars with antibodies for ASFV [35]. This report from Gallardo et al. [35] agrees with a previous work performed by Fernández-Pinero et al. [26], in which it was proved that a PCR with a Universal Probe Library (UPL) was more sensitive than the early WOAH-PCR recommended test developed by King et al. [22], since lower Ct values were obtained for 20 ASFV reference isolates. It was concluded that the robustness of the method developed by King et al. [22] decreased when positive samples with a weak signal were tested [26]. Hence, the obtained results during the validation of the ASFV MONODOSE dtec-qPCR are supported by previous published works. These data enhance the importance of keeping up-to-date the recommended ASFV qPCR designs in order to enable the detection of all possible circulating ASFV genotypes and face all possible epidemiological scenarios.

The ASFV MONODOSE dtec-PCR kit incorporates additional benefits, such as individual ready-to-use tubes which include all the components needed for the specific detection of this pathogen by performing a qPCR test, enabling the minimisation of time invested in reaction preparation. This technology is very user-friendly and straightforward, as all the reagents required are dehydrated together, which enables technicians to just add their samples and run the PCR without requiring any intermediate steps, therefore avoiding possible mistakes and incidences derived from a long manipulation process. Another benefit granted by this kit format is that it can be transported at room temperature, avoiding the use of dried ice, considerably reducing the required time for delivery and shipment expenses. The risk of enzyme rupturing has been eliminated because the enzyme is not frozen–thawed. Also, cross-contamination and fluorophore deterioration by UV light have been minimised to the greatest extent possible due to the dehydration of the reagents and the reduction in manipulation time required in the MONODOSE format. This product was registered in the Registry of Entities and Animal Health Products under the Ministry of Agriculture, Fisheries and Food of Spain (MAPA) with the following references: entity authorisation number R-10046 and ASFV dtec-qPCR registration number 11033-RD.

## 5. Conclusions

In the present article, a new qPCR previously designed for the detection of African swine fever virus, the ASFV MONODOSE dtec-qPCR kit developed by GPS™ and fully available worldwide, was validated following the guidelines recommended by the international standard UNE-EN ISO/IEC 17025:2005. The qPCR method showed a 100% diagnostic sensitivity, specificity and efficiency in this study when it was used with different porcine tissue samples doped with reference ASFV genetic material. These findings suggest that the method is an effective, fast, sensitive and specific tool for the detection of ASFV in clinical porcine samples. Furthermore, the newly developed kit was compared with reference qPCR methodologies endorsed by the WOAH. Data obtained agreed with the most recently developed method, indicating that this kit may be confidently used for the detection of ASFV in porcine samples. Also, the results highlighted the need to update the existing technology, as publicly available data are constantly increasing and further improvements can still be achieved.

This kit, with full analytical and diagnostic validation following international standards, improves the results obtained with most of the current reference methodologies and brings additional benefits to qPCR technology.

## Figures and Tables

**Figure 1 vetsci-10-00564-f001:**
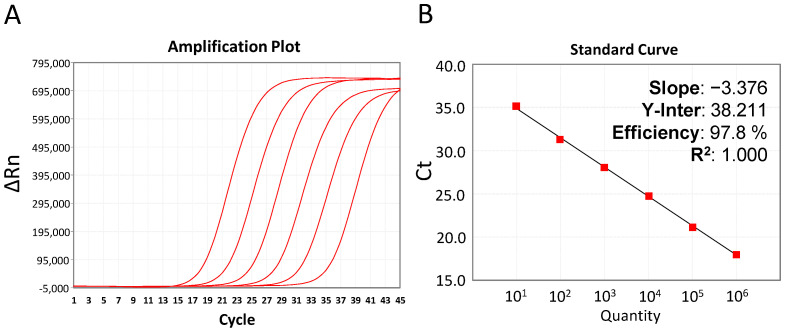
Quality control of the ASFV MONODOSE dtec-qPCR kit with data of six ranges of decimal dilution from 10^6^ copies to 10 copies of the Standard Template provided in the kit, and negative control. (**A**) Amplification plot and (**B**) a representative calibration curve with stats.

**Figure 2 vetsci-10-00564-f002:**
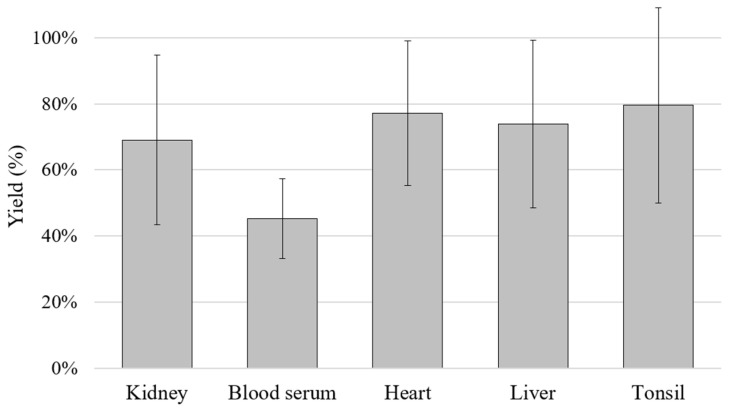
Quality control of the ASFV MONODOSE dtec-qPCR kit with data of six ranges of decimal dilution from.

**Table 1 vetsci-10-00564-t001:** Summarised values, criteria for acceptance and results obtained with the ASFV MONODOSE dtec-qPCR kit. n, repetitions for each parameter.

Term of Validation	Obtained Values		Acceptance Criteria	Conclusion
Specificity	Inclusivity: in silico assays against all available sequences from the target species. Assessed with 21 reference DNAs (INIA-CISA) and 8 reference samples (Poland) positive for ASFV		Inclusivity: positive amplification for all ASFV strains and for all reference samples	ACCEPTED
	Exclusivity: in silico studies against all available sequences		Exclusivity: negative amplification for virus other than ASFV	ACCEPTED
Standard curve n = 10	Y = −3.369 × X + 36.375 a = −3.369 R^2^ = 1.000		−4.114 < a < −2.839 R^2^ > 0.98	ACCEPTED
	F_assay_ = 3.261 F_fisher_ = 5.318		F_assay_ < F_fisher_	ACCEPTED
	Efficiency (*e*) = 98.1%		75% < *e* < 125%	VALIDATED
Reliability n = 10	Repeatability		CV < 10%	REPEATABLE
	Conc.	CV (%)		
	10^6^ copies	1.94%		
	10^5^ copies	0.65%		
	10^4^ copies	0.62%		
	10^3^ copies	0.63%		
	10^2^ copies	0.90%		
	10^1^ copies	1.12%		
	Reproducibility		CV < 10%	REPRODUCIBLE
	Conc.	CV (%)		
	10^6^ copies	1.65%		
	10^5^ copies	0.95%		
	10^4^ copies	0.97%		
	10^3^ copies	0.86%		
	10^2^ copies	1.04%		
	10^1^ copies	1.34%		
Detection limit (LOD) n = 15	10 copies	Posit = 15/15	Positives ≥ 90%	ACCEPTED
Quantification limit (LOQ) n = 15	10 copies	*t* value = 1.161	*t* value < *t*_student_	ACCEPTED
		*t*_student_ = 2.145		
Diagnostic sensitivity	True Positives: 105 False Negatives: 0 SD = 100%		DS > 90%	ACCEPTED
Diagnostic specificity	True Positives: 76 False Positives: 0 ED = 100%		DE > 90%	ACCEPTED

**Table 2 vetsci-10-00564-t002:** Characteristics of the reference ASFV genomic material and results obtained with the ASFV MONODOSE dtec-qPCR Test.

Isolate	Origin Country	Host	Year	Town/Province	Geno Type	Laboratory	GPS™ KIT (Ct)
E70	Spain	Pig	1970	Pontevedra	I	INIA-CISA *	24.73
BF07 OUAGA 2	Burkina	Pig	2007	Ouaguodaga	I	INIA-CISA *	27.49
SS14/WB-Sassari1	Italy	Wild boar	2014	Sassari	I	INIA-CISA *	25.82
SS14/DP-Cagliari1	Italy	Pig	2014	Cagliari	I	INIA-CISA *	29.90
Arm07	Armenia	Pig	2007	Dilijan	II	INIA-CISA *	24.94
Ukr12/Zapo	Ukraine	Pig	2012	Zaporozhye	II	INIA-CISA *	23.87
Ukr15/DP-Kieve 1	Ukraine	Pig	2015	Kiev	II	INIA-CISA *	26.73
LT14/1490	Lithuania	Wild boar	2014	Vilnius	II	INIA-CISA *	24.16
Pol14/Krus	Poland	Wild boar	2014	Podlaskie	II	INIA-CISA *	24.96
Lv14/DP/Robez3	Latvia	Pig	2014	Dienvidlatgale	II	INIA-CISA *	27.50
Est14/WB-Valga-1	Estonia	Wild boar	2014	Valga	II	INIA-CISA *	27.59
Est15/WB-Tartu14	Estonia	Wild boar	2015	Tartu	II	INIA-CISA *	29.16
MOL16/DP-CERNO1	Moldova	Pig	2016	Cernoleuca	II	INIA-CISA *	28.35
MOL16/DP-MOSA1	Moldova	Pig	2016	Mosana	II	INIA-CISA *	25.96
Moz64	Mozambique	Pig	1964	NK	V	INIA-CISA *	24.58
MwLil 20/1	Malawi	Tick	1983	Chalaswa	VIII	Complete genome	26.34
Ken11/KisP52	Kenya	Pig	2011	Kisumu	IX	INIA-CISA *	28.42
Ken06.Bus	Kenya	Pig	2006	Busia	IX	INIA-CISA *	30.54
Ken08Tk.2/1	Kenya	Tick	2007	Kapiti	X	INIA-CISA *	27.40
UG10/Tk3.2	Uganda	Tick	2010	Mburu	X	INIA-CISA *	32.35
Eth13/1505	Ethiopia	Pig	2013	Bishoftu	XXIII	INIA-CISA *	26.51
Pol18/Var1	Poland	Pig	2018	Warsaw	-	PIWet **	24.73
Pol18/Var2	Poland	Pig	2018	Warsaw	-	PIWet **	27.49
Pol18/Var3	Poland	Wild boar	2018	Warsaw	-	PIWet **	25.82
Pol18/Var4	Poland	Wild boar	2018	Warsaw	-	PIWet **	29.90
Pol18/Var5	Poland	Wild boar	2018	Warsaw	-	PIWet **	24.94
Pol18/Var6	Poland	Wild boar	2018	Warsaw	-	PIWet **	23.87
Pol18/Var7	Poland	Pig	2018	Warsaw	-	PIWet **	26.73
Pol18/Var8	Poland	Pig	2018	Warsaw	-	PIWet **	24.16

* INIA-CISA, Centro de Investigación en Sanidad Animal del Instituto Nacional de Investigación y Tecnología Agraria y Alimentaria, Spain. ** PIWet, Państwowy Instytut Weterynaryjny, Poland.

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
