# Peer review of "Internal Validation of the ASFV MONODOSE dtec-qPCR Kit for African Swine Fever Virus Detection under the UNE-EN ISO/IEC 17025:2005 Criteria"

_vetsci, 2023, doi:10.3390/vetsci10090564_

Round 1
Reviewer 1 Report
The present manuscript reports the validation of a new qPCR method for the detection of African Swine Fever Virus, that is easier to use and has a faster protocol than the current recommended tests for ASFV detection. The manuscript is overall very well written and detailed, but some points need to be clearer for the readers, as detailed below. The two biggest concerns raised while reviewing this manuscript are: 1) How do the authors claim to have developed a diagnostic method that is 100% sensitive and 100% specific, as it is scientifically impossible to obtain such results? 2) The samples used for the validation of the test were artificially positive, as they were doped with pure genetic material; How would be the sensitivity and specificity results if naturally positive samples (clinical samples) were used? Without acknowledging these limitations, the manuscript misleads the readers to believe that the methods used for comparison are much inferior than the proposed method, which it was not proven by the data presented. Ideally, clinical samples need to be included in the tests to improve the analysis of sensitivity and specificity.
Page 1, Simple summary and Abstract: It is not clear to me why the authors are mentioning the ASFv infection only in boars, since both feral and domestic swine are susceptible. If there is a specific reason for mentioning only boars in the background, it needs to be clearly acknowledged.
Page 1, line 21: While viruses are naturally multidrug resistant microorganisms, the sentence is misleading, as this expression is usually used for bacterial pathogens. I suggest to re-write it in a more instructive way. Suggestion: “ASF virus is considered an emerging virus that that causes African swine fever, a disease characterized by high mortality and elevated transmission rates and that, as at it is for most other viral diseases, cannot be treated with specific drugs.”
Page 1, lines 33-34: What are these technical innovations and benefits? Make it clear, as such improvements will be extremely important to users.
Page 3, line 118: Which sample? Pure ASF reference genetic material or the porcine samples doped with this reference ASF genetic material?
Page 3, lines 112-123: What was used as positive control? And for negative control? What was the external target mentioned as Internal Control? – Readers need more details to be able to reproduce the methodology.
Page 3: What were the primers/targets for each PCR method?
Page 7, table 2: Replace “board” by “boar”
Methods:
- If the authors have access to ASFv biological positive samples, they should be tested with the proposed method instead of using only DNA doped samples.
- It would be interesting to see meat samples tested as one doped matrix.
Results:
- If the ready-to-use tubes are considered by the author as one of the biggest advantages of the proposed method, include at least one (supplementary) figure of the tubes.
Discussion:
- How do the authors explain a test with 100% sensitivity and specificity, as such results are scientifically impossible to reach?
- Page 10, lines 329-332: The present study did not test samples obtained from infected swine, so, such statement should be removed from the discussion as it is detrimental do the other publication. If the authors decide to keep the sentence, they need to make it clear that such test was not performed with the proposed protocol, therefore, it is not possible to make a direct comparison.
- Page 11, lines 353-354: How the deterioration by UV light was optimized?
No additional comments.
Reviewer 2 Report
This study validated a PCR test for ASFV by assessing repeatability, sensitivity, specificity, etc. Although the data was presented with much details, Key points in terms of assay development include:
1. Detect all strains. Please list the target gene of this test. The target gene would be very helpful to assess the assay performance if divergent strains emerge.
2. The diagnostic performance was evaluated using spiked porcine samples. In another word, it does not reflect the real performance for testing field samples.
Round 2
Reviewer 1 Report
With the clarifications and modifications made by the authors, the manuscript is now suitable for publication.